# Effect of Dietary Standardized Ileal Digestible Arginine to Lysine Ratio on Reproductive Performance, Plasma Biochemical Index, and Immunity of Gestating Sows

**DOI:** 10.3390/ani14182688

**Published:** 2024-09-15

**Authors:** Xiaolu Wen, Zongyong Jiang, Xuefen Yang, Hao Xiao, Kaiguo Gao, Li Wang

**Affiliations:** Institute of Animal Science, Guangdong Academy of Agricultural Sciences, State Key Laboratory of Swine and Poultry Breeding Industry, Key Laboratory of Animal Nutrition and Feed Science in South China, Ministry of Agriculture and Rural Affairs, Guangdong Provincial Key Laboratory of Animal Breeding and Nutrition, 1 Dafeng 1st Street, Guangzhou 510640, China; wenxiaolu@gdaas.cn (X.W.); jiangzy@gdaas.cn (Z.J.); yangxuefen@gdaas.cn (X.Y.); xiaohao@gdaas.cn (H.X.)

**Keywords:** Arg to lysine ratio, gestating sow, reproduction performance, colostrum composition, immunity

## Abstract

**Simple Summary:**

Studies have shown that dietary arginine supplementation can promote placental growth and vascular development in pregnant sows, increase litter size and improve the reproductive performance of sows. However, there are few studies on the optimal arginine and arginine to lysine ratio in sows during pregnancy. In this study, we investigated the optimal standardized ileal digestible arginine to lysine (SID Arg: Lys) ratio on the reproductive performance, immune response, and biochemical parameters of sows during gestation, as well as the colostrum composition and performance of their offspring. Our results indicate that an appropriate dietary SID Arg: Lys ratio can increase the number of piglets born alive, reduce the birth interval, and promote milk fat synthesis.

**Abstract:**

The aim of this study was to determine the optimal SID Arg: Lys ratio for maximizing the reproductive performance, immunity and biochemical parameters of sows during gestation, the colostrum composition, and the performance of their offspring. A total of 174 multiparous sows were randomly allocated to five treatment groups varying in dietary SID Arg: Lys ratios (0.91, 1.02, 1.14, 1.25 and 1.38) through modification of the levels of Arg or alanine supplementation (the total level of nitrogen was the same in all treatments). The results showed that increasing the dietary SID Arg: Lys ratio increased the number of piglets born alive (*p* < 0.05, linear and quadratic). The number of stillborn piglets, the birth weight variation of born alive piglets, the birth interval (*p* < 0.05, linear and quadratic) and the number of mummies (*p* < 0.05, quadratic) reduced with increasing the dietary SID Arg: Lys ratio. Broken-line regression analysis indicated that the optimal SID Arg: Lys ratio requirement for gestating sows to maximize the number of piglets born alive was 1.25. The content of non-fat solid, total solid, protein, and energy in colostrum increased linearly and quadratically (*p* < 0.05) with increasing dietary SID Arg: Lys ratio. Additionally, when increasing the dietary SID Arg: Lys ratio, the concentration of IgA (*p* < 0.05, quadratic) and IgM (*p* < 0.05, linear and quadratic) of plasma increased at day 90 of gestation; IgG (*p* < 0.05, linear and quadratic) concentration increased at day 110 of gestation of sows. The dietary SID Arg: Lys ratio had an increasing effect (*p* < 0.05, linear and quadratic) on plasma insulin levels at day 90 of gestation. Furthermore, there were increases in plasma concentrations of nitric oxide and ornithine at day 110 of gestation, Arg at day 90 and 110 of gestation (*p* < 0.05, linear and quadratic) and total protein at day 110 of gestation (*p* < 0.05, linear) with increasing dietary SID Arg: Lys ratio. The results of our study indicated that increases in the dietary SID Arg: Lys ratio during gestation resulted in an increase in the number of piglets born alive, a decrease in birth intervals, and an improvement in immunity and colostrum composition. The optimal SID Arg: Lys ratio for gestating sows to maximize the number of piglets born alive was 1.25.

## 1. Introduction

Arg is an essential functional amino acid for gestating sows, and it is important for the development of the placenta and fetus [1]. In addition to participating in protein synthesis, Arg also serves as a signaling molecule to promote protein synthesis in embryo and uterus, mainly by activation of the mammalian target of the rapamycin pathway [2,3]. Dietary supplementation with Arg in gestating sows has been shown to significantly enhance the total number of born piglets and litter weight and reduce the number of stillbirths [4,5]. The transporters in the intestine that mediate the absorption of lysine also mediate the absorption of Arg, which is a cationic amino acid [3]. Arg might thus compete with lysine for absorption, transport and metabolism. Maintaining a proper Arg to lysine ratio is critically important for maintaining the reproductive performance of sows. The recommended SID Arg: Lys ratio of gestating sows with more than three parities was 0.51–0.53 [6]. It was reported that increasing the level of Arg in the diet significantly improved the reproductive performance of sows [7]. However, few recent studies have focused on the effect of SID Arg: Lys ratios during gestation on the reproductive performance of sows. Therefore, the purpose of this study was to investigate the effect of increasing dietary SID Arg: Lys ratios on the reproductive performance, immunity and biochemical parameters of gestating sows and the colostrum compositions and litter performance, as well as to evaluate the optimal dietary SID Arg: Lys ratio for gestating sows.

## 2. Materials and Methods

### 2.1. Swine and Diet

The experimental protocol was approved by the Animal Care Committee of the Institute of Animal Science, Guangdong Academy of Agriculture Science, Guangzhou, P. R. China, with an approval number of GAASISA-2017-028.

The study was performed on a commercial farm with 2000 sows located in Guangxi province of China. A total of 174 multiparous sows (Landrace × Large white; 3~6 parity) with an initial body backfat thickness of 18.20 ± 1.43 mm (measured at weaning day) and previous litter size of 12.51 ± 0.25 were equally allocated into five groups (n = 36, 34, 35, 36, 33, respectively) based on expected farrowing date, parity, backfat thickness and prior success in gestating and fed diets with different SID Arg: Lys dietary ratios of 0.91, 1.02, 1.14, 1.25 and 1.38. The experimental diet was administered at day 30 of gestation. The five dietary treatments were formulated by supplementation with 0, 0.08, 0.16, 0.24, and 0.32 g/kg Arg, respectively, and maintained constant levels of lysine (average SID Lys 0.64%) and threonine (average SID Arg 0.58%) among groups (Table 1). Additionally, all diets (Table 1) were formulated to be isocaloric (2300 kcal NE/kg) and contained the same nutritional compositions, which exceeded the nutrient requirement recommendations of NRC 2012 for sows except the Arg level. Animals were housed individually in gestation crates (2.1 × 0.7 m) before they were moved to the farrowing room and fed twice each day at 7:30 and 16:30 with a total amount of 2.3 kg/d from days 0 to 30 of gestation, 2.5 kg/d from days 31 to 90 of gestation, and 2.8 kg/d from day 90 of gestation to farrowing.

### 2.2. Recording and Sampling 

The backfat thickness of all sows was measured at days 30 and 110 of gestation using an ultrasonic device (Agroscan A16, Echo Control Medical, Angouleme, France) at the P2 position (6 cm from the midline at the head of the last rib). After farrowing, the numbers of total born, born alive, stillborn and mummified piglets were recorded. Piglet weight was recorded individually at farrowing. The uniformity of newborns was determined by intralitter coefficient of variation (CV), as calculated via dividing the standard deviation by the average birth weight within the litter. Thirty sows (six sows each group) were randomly selected for blood sampling. On days 90 and 110 of gestation, Heparinized blood samples (10 mL) were collected from the ear vein 2 h after feeding and centrifugated at 1000× *g* for 10 min at 4 °C to harvest the plasma samples followed by storage at −80 °C until further analysis of amino acids, plasma biochemical indices and immunoglobulin. Within 2 h of the first piglet’s delivery, colostrum was collected from all functional teats after properly cleansing the udder with water. Approximately 30 mL of colostrum was collected into Falcon tubes and stored at −20 °C until analysis for milk composition. 

### 2.3. Chemical Analyses

Samples of the basal diet were analyzed for the contents of dry matter (DM) and crude protein according to AOAC (2000) methods [8]. Plasma concentrations of free amino acids were measured in acid hydrolysates with an analyzer (L-8900, Hitachi, Tokyo, Japan), as described by Ma et al. [9]. The contents of protein, lactose, fat and non-fat solids in colostrum were measured by the Julie Z7 automatic analyzer (Scope Electric, Regensburg, Germany). Energy concentration in colostrum was calculated using the following energy values: 39.8 kJ/g fat, 23.9 kJ/g protein and 16.5 kJ/g lactose [10]. The concentrations of urea nitrogen (UN) and nitric oxide (NO) in plasma were measured with commercial kits according to the manufacturer’s instructions (Nanjing Jiancheng Biotechnology Co. Ltd., Nanjing, China). Biochemical parameters (total protein, albumin, uric acid, glucose, triacylglycerol, cholesterol, HDL-cholesterol, LDL-cholesterol, calcium and phosphorus) in plasma were analyzed with the automatic biochemical analyzer (Selectra Pro XL, Vital Scientific, Spankeren, Gelderland, The Netherlands). An enzyme-linked immunosorbent assay (Nanjing Jiancheng Bioengineering Institute Co., Ltd., Nanjing, China) was performed to detect plasma immunoglobulin A (IgA), immunoglobulin G (IgG) and immunoglobulin M (IgM) following the recommended protocol by the manufacturer. Pig ELISA kits (CUSABIO Biotech Co., Ltd., Wuhan, China) were used to measure plasma concentrations of progesterone, growth hormone (GH), insulin, Insulin-Like Growth Factor-1 (IGF-1) and estrogen, following the manufacturer’s instructions.

### 2.4. Statistical Analysis

Statistical analyses were conducted using the PROC MIXED procedure of SAS 9.2 (SAS Inst. Inc., Cary, NC, USA), with each litter as an experimental unit. The reproductive cycle and dietary SID Arg: Lys ratio were specified as fixed effects. In the mixed model, the response variables included the number of born alive piglets, litter weight of born alive piglets, backfat thickness, placental weight, urea nitrogen concentration, colostrum compositions, and AA concentration. Regression analyses were performed to evaluate the linear and quadratic effects of the dietary SID Arg: Lys ratio. Data obtained by ANOVA are shown as the mean and SEM. Statistical significance was declared at *p* < 0.05. Multiple comparisons were made when the ANOVA indicated significant differences. Tukey’s test was used in multiple comparisons of means to adjust the *p*-values when using a mixed model procedure for data analysis. Straight broken-line and quadratic broken-line regression analyses were used to estimate the SID Arg: Lys requirements for sows by using the NLIN procedure in SAS 9.2 [11].

## 3. Results

### 3.1. Reproductive Performance of Sows

The effects of dietary SID Arg: Lys ratios on the reproductive performance of sows are shown in Table 2. The number of piglets born alive and litter weight increased linearly (*p* < 0.05) and quadratically *(p* < 0.05), stillborn size decreased linearly (*p* < 0.05) and quadratically (*p* < 0.05), and the number of mummified piglets decreased quadratically (*p* < 0.05) as the dietary SID Arg: Lys ratio increased. The CV of total born birth weight decreased linearly (*p* < 0.05) and quadratically (*p* < 0.05) as the dietary SID Arg: Lys ratio increased. Moreover, the straight broken-line and quadratic broken-line regression analysis indicated that the SID Arg: Lys ratio for maximum number of piglets born alive was 1.25 and 1.39 (Figure 1). For litter weight of born alive piglets, the break-points were 1.25 and 1.44 for the linear and quadratic models, respectively (Figure 2). According to feed intake and Arg concentration in feed, the daily Arg requirement of gestating sows is 20 g/day (<90 days) and 24 g/day (>90 days). There were no differences in total born litter size or average live birth weight (*p* > 0.05) among the five treatment groups.

The farrowing duration of sows fed a diet with an SID Arg: Lys ratio of 0.91 was higher than that of sows fed diets with SID Arg: Lys ratios of 1.02, 1.14 and 1.38 (Table 3, *p* < 0.05). Birth intervals decreased linearly (*p* < 0.05) and quadratically (*p* < 0.05) with increasing the dietary SID Arg: Lys ratio. Birth intervals of sows fed a diet with an SID Arg: Lys ratio of 0.91 were higher than that in other groups (*p* < 0.05). Furthermore, dietary SID Arg: Lys ratio had no significant effect on backfat thickness and backfat gain during days 30 to 110 in sows (Table 4, *p* > 0.05).

### 3.2. Colostrum Composition

The effects of dietary SID Arg: Lys ratio on the composition of colostrum are shown in Table 3. There was a quadratic effect of increasing the SID Arg: Lys ratio on the percentage of milk fat (*p* < 0.05), non-fat solid (*p* < 0.05), total solid (*p* < 0.05), milk protein (*p* < 0.05) and energy (*p* < 0.05). The dietary SID Arg: Lys ratio also had a linear effect on the concentrations of non-fat solid (*p* < 0.05), total solid (*p* < 0.05), milk protein (*p* < 0.05) and energy (*p* < 0.05) in the colostrum, with highest non-fat solid (*p* < 0.05), total solid (*p* < 0.05), milk protein (*p* < 0.05) and energy levels in the colostrum of sows fed diets with an SID Arg: Lys ratio of 1.25. Moreover, the percentage of lactose in the colostrum of sows fed diets with an SID Arg: Lys ratio of 0.91 was higher than that in the 1.02 and 1.25 ratio groups (*p* < 0.05).

### 3.3. Plasma IgA, IgG, IgM

The concentration of IgA in plasma increased quadratically (*p* < 0.05) as the dietary SID Arg: Lys ratio increased at day 90 of gestation (Table 4). The IgG concentration increased linearly (*p* < 0.05) and quadratically (*p* < 0.05) as the SID Arg: Lys ratio increased at day 110 of gestation. The IgM concentration increased linearly (*p* < 0.05) and quadratically (*p* < 0.05) as the SID Arg: Lys ratio increased at day 90 of gestation.

### 3.4. Plasma Biochemical Indexes and Hormones

No effects of dietary SID Arg: Lys ratio on the plasma urea nitrogen, IGF-1, progesterone or estrogen of sows at days 90 or 110 of gestation were found (Table 5). The concentration of NO in plasma was increased linearly (*p* < 0.05) and quadratically (*p* < 0.05) as the SID Arg: Lys ratio increased at day 110 of gestation. The concentration of NO in plasma in the 0.91 ratio group was lower than that in other groups at day 110 of gestation (*p* < 0.05). On day 90 of gestation, the plasma insulin level increased linearly (*p* < 0.05) and quadratically (*p* < 0.05) with an increased SID Arg: Lys ratio in the gestating diet. Moreover, the concentrations of total protein in plasma increased linearly (*p* < 0.05) as the SID Arg: Lys ratio increased at day 110 of gestation (Table 6). There were no significant differences in the plasma concentrations of albumin, creatinine, uric acid, glucose, triacylglycerol, cholesterol, HDL-cholesterol, LDL-cholesterol, calcium and phosphorus among groups (*p* > 0.05).

### 3.5. Plasma Concentrations of Free Amino Acids

Plasma tyrosine, ornithine and Arg concentrations increased linearly (*p* < 0.05) and quadratically (*p* < 0.05), while plasma concentrations of glutamate decreased linearly (*p* < 0.05), and concentrations of threonine increased quadratically (*p* < 0.05) as dietary SID Arg: Lys ratio increased at day 90 of gestation (Table 7). The plasma methionine concentration of the 1.25 group was significantly lower than that of the 1.14 and 1.38 group at day 90 of gestation (*p* < 0.05). The concentration of taurine, glutamine, methionine, isoleucine, lysine and Arg increased linearly (*p* < 0.05) and quadratically (*p* < 0.05), and concentrations of leucine increased quadratically (*p* < 0.05) as the dietary SID Arg: Lys ratio increased at day 110 of gestation (Table 8). Moreover, the plasma alanine concentration decreased linearly (*p* < 0.05) and quadratically *(p* < 0.05) as the dietary SID Arg: Lys ratio increased at day 90 and 110 of gestation. There were no differences in the concentrations of the other free amino acids in plasma among groups (*p* > 0.05)

## 4. Discussion

Recent studies have reported that increased dietary Arg: Lys ratios during gestation improved the reproductive performance of sows, increased the number of piglets born alive [12] and litter weight [4], and reduced the number of stillborn piglets [13]. The purpose of this study was to determine the optimal SID Arg: Lys ratio for gestating sows. Dietary amino acid levels were based on NRC (2012) recommendations and production in commercial practice with SID lysine set at 0.64%, and we gradually increased the dietary SID Arg: Lys ratios (0.91 to 1.38) by supplementing with different levels of Arg (0 to 0.32%). In the current study, the straight broken-line and quadratic broken-line regression analysis indicated that the SID Arg: Lys ratio for the maximum number of born alive piglets was 1.25 and 1.39, whereas the maximum weight of born alive piglets was 1.25 and 1.44. The findings from this study suggest that the actual daily SID Arg requirement for gestating sows (3th parity) may be higher than 0.28%, which was the NRC (2012) recommendation. When the dietary SID Arg: Lys ratio increased from 0.91 to 1.25, the number of piglets born alive was significantly increased, while the number of stillborn piglets was reduced. The decrease in the number of stillborn piglets may be attributed to an increase in the Arg metabolite NO, which expands the uterus [14,15]. The maximum amount of arginine added in this study was 0.32%, and the maximum SID Arg: Lys ratio was 1.38. The recommendation value of the quadratic broken-line model was higher than 1.38. It is suggested that the number of piglets born alive may be increased when the amount of arginine added continues to increase. An increase in the ratio of SID Arg: Lys had no effect on the average live birth weight but reduced the CV of total born birth weight. This may be because Arg promotes the growth of low-weight fetuses [16]. The longer farrowing duration has been shown to make the sows lose extra physical energy and produce tremendous creatine during farrowing, which extends the postnatal recovery of sows and increases the mortality of newborn piglets due to the increase in hypoxia in piglets [17,18,19]. The reasons for the long farrowing duration mainly include the large number of total born, the lack of exercise in sows, and the decrease in oxytocin concentration [20]. This study showed that increasing the proportion of dietary SID Arg: Lys ratio significantly reduced the birth interval and farrowing duration of piglets potentially by promoting the uniformity of piglets. However, some previous studies showed that an increased dietary Arg: Lys ratio did not influence the reproduction of sows [21,22]. Other studies have not found a benefit of Arg supplementation, which may be due to the short duration of Arg supplementation [21] and the low level of lysine in the diet, resulting in an imbalance between Arg and lysine.

Backfat reflects the body condition and energy reserve of sows at different physiological stages, and the thickness of backfat is often used as one of the most important indicators to reflect nutritional status and strongly relates to reproductive performance of sows [23,24]. It is reported that sows with proper backfat thickness had better litter performance, birth interval, mating success rate and conception delivery rate [25,26,27,28]. In this study, the proportion of dietary SID Arg: Lys ratio had no significant effect on the backfat thickness and backfat gain of sows during gestation. Similarly, a previous study also found that daily Arg supplementation at 25.5 g in sow diets from the 77th day of pregnancy had no significant effect on the backfat thickness of sows [16]. Mateo et al. also found that adding 1% Arg-HCl from 30 days of pregnancy had no effect on the backfat thickness of sows at 70, 90 and 110 days of gestation [15].

The growth performance and early mortality of piglets are related to milk production and compositions of nutrients in milk, mainly including protein, fat and lactose [29,30]. Colostrum contains a large number of immunoglobulins (IgG, IgA, IgM), which can provide immediate immune protection for newborn piglets [13]. A previous study has shown that dietary nutritional approaches affect the concentration of nutrients in colostrum [31,32]. In this study, the concentration of milk fat in colostrum was increased quadratically with an increase in the dietary SID Arg: Lys ratio in gestating sows. Milk fat plays an important role in the regulation of piglet body temperature and affects the early survival rate of piglets [33]. It has been demonstrated that an increase in milk fat might be related to a decrease in lactose concentration, thus contributing to a significant increase in piglet body weight during lactation [34]. Furthermore, the concentration of total protein, non-fat solid, total solid and energy in colostrum increased significantly with the increase in the dietary SID Arg: Lys ratio of gestating sows. Arg may be a limiting factor in colostrum synthesis [35]. Krogh et al. reported that the concentration of milk protein and DM in colostrum significantly increased by adding Arg to pregnancy diet, which was consistent with our results [10].

Arg improves the immune performance of pigs [36], activates T cells, and promotes maturation and development of B cells [37]. Previous studies have found that adding Arg to the diet of gestating sows significantly increases the concentration of IgG and IgM in plasma [12]. Moreover, supplementation of Arg in the diet of sows at late gestation can significantly increase the concentration of immunoglobulin G in colostrum and increase the birth weight of piglets [13]. We also found that increasing the dietary SID Arg: Lys ratio significantly increased the plasma immunoglobulin concentration. This finding indicates that an appropriate SID Arg: Lys ratio is critically important for improving immune performance. Metabolites such as NO and polyamine produced during Arg metabolism not only stimulate placental angiogenesis and blood vessel growth but also promote the transfer of uterus placenta blood flow and maternal nutrients [38]. All these effects contribute to the development of effective uterine capacity for fetal growth and development [39,40]. This study also showed that increasing the dietary SID Arg: Lys ratio significantly increased the concentration of NO in the plasma of sows at day 110 of gestation. This finding was consistent with previous research [41]. Nitric oxide synthase metabolizes Arg, producing NO [42]. Insulin is an important physiological hormone that not only regulates glucose and lipid metabolism but also plays an important role in the reproductive performance of sows. Growth hormone supports fetal growth, and there is a strong positive association between insulin and growth hormone [43], so a rise in insulin may be associated with fetal birth weight. Arg has been shown to stimulate the release of insulin in pigs [44]. This study found that increasing the dietary proportion of Arg significantly increased the plasma insulin concentration. Arg is a cationic amino acid that stimulates β cells to secrete insulin through membrane depolarization [45]. Progesterone and estrogen play important roles in mammalian pregnancy maintenance. Previous studies have found that Arg stimulates the secretion of estrogen [4]. This may be due to the relatively low amount of Arg added to the diet in this study. We also found that increasing the dietary Arg: Lys ratios during gestation had no significant influence on the plasma concentrations of urea nitrogen, albumin, creatinine, uric acid, glucose, triacylglycerol, cholesterol, HDL-cholesterol, LDL-cholesterol, calcium and phosphorus, which is consistent with the previous research results on urea nitrogen [21]. Additionally, plasma total proteins, including albumin, globulin and fibrin, which can reflect the strength of protein synthesis, metabolism and immune status. A higher plasma total protein content might promote growth and improve the feed conversion efficiency of animals. In this study, the concentrations of total protein in plasma increased linearly as the dietary SID Arg: Lys ratio increased, indicating that a high dietary SID Arg: Lys ratio during gestation might enhance the protein synthesis ability of sows.

The results of free amino acid in plasma showed that the concentrations of ornithine and Arg increased linearly by increasing the dietary SID Arg: Lys ratio at day 90 of gestation. These results are consistent with previous studies [4,5,12]. Ornithine is a metabolite of Arg and acts as a necessary precursor for the synthesis of NO and polyamine [46]. Enhanced synthesis of NO from Arg may mediate placental angiogenesis and placental vascular growth, so as to promote the growth and development of the fetus [47]. However, we also observed that increasing the dietary SID Arg: Lys ratio of gestating sows increased plasma concentrations of glutamine, methionine, isoleucine and lysine linearly and quadratically at day 110 of gestation and increased plasma concentrations of threonine and tyrosine at day 90 of gestation. It is possible that dietary Arg promotes the absorption and synthesis of amino acids, thereby increasing the concentration of protein in plasma. Excessive Arg reduces the absorption of lysine in pigs [48]. But the dietary SID Arg: Lys ratios of 0.91 to 1.38 used here had a positive effect on free lysine in sow plasma, which is inconsistent with previous results [15]. This may be attributed to the relatively low Arg requirement of piglets and the high doses of Arg added (0.5–2.0%) [48], which exceed the requirement of piglets, thus inhibiting the absorption of lysine. However, in this study, the Arg requirement for sows during pregnancy was relatively high, and the amount of Arg added in this test was comparatively low (0.08–0.24%), which did not inhibit the absorption of lysine.

## 5. Conclusions

The current study indicated that increasing the dietary SID Arg: Lys ratio (0.91~1.38) during gestation increased the number of born alive, which might be associated with an improvement in immunity, increased levels of NO and ornithine in plasma and a decreased birth interval. According to the straight broken-line model, the indicated dietary SID Arg: Lys ratio for sows during gestation to maximize the number of born alive piglets was 1.25.

## Figures and Tables

**Figure 1 animals-14-02688-f001:**
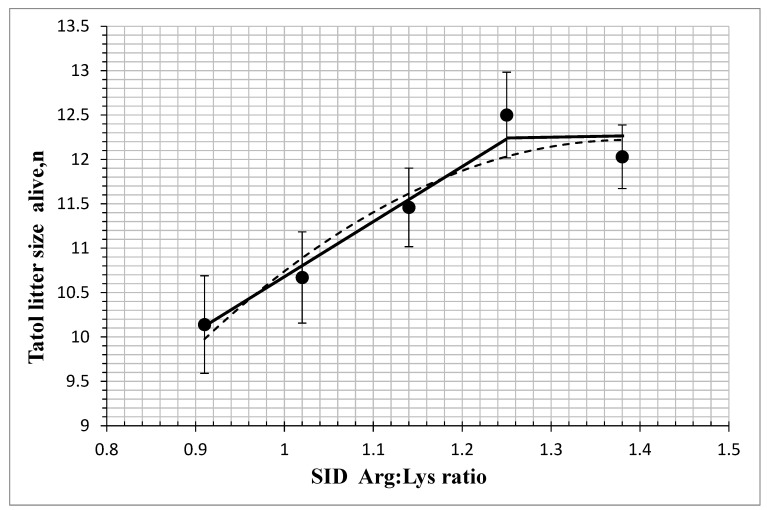
Fitted straight broken-line and quadratic broken-line regression models on the number of born alive piglets as a function of increasing standardized ileal digestible (SID) Arg: Lys in gestating sows. The optimum SID Arg: Lys ratio determined by straight broken-line model was 1.25 [Y = 12.181 − 6.1628 (1.25-SID Arg: Lys ratio) (closed line); R^2^ = 0.963, *p* < 0.05]. Using the quadratic broken-line model, the optimum SID Arg: Lys ratio was 1.39 [Y = 12.2198 − 9.8062 × (1.39 − SID Arg: Lys)^2^ (open line); R^2^ = 0.895, *p* < 0.05]. Data points (●) represent treatment means (n = 35, 34, 35, 36, 33).

**Figure 2 animals-14-02688-f002:**
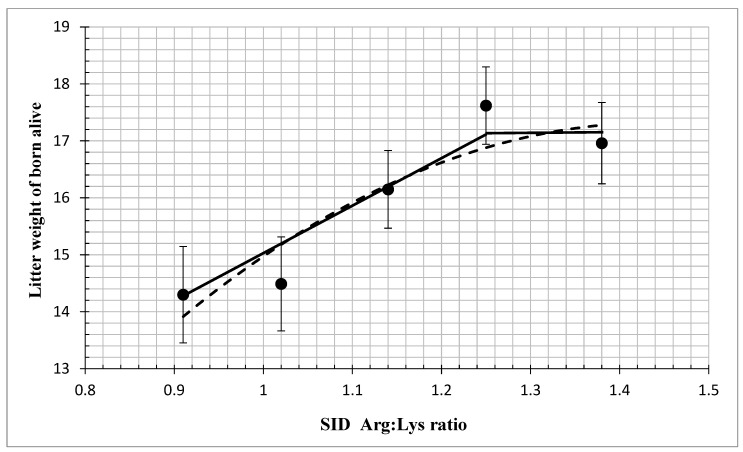
Fitted straight broken-line and quadratic broken-line regression models on the litter weight of born alive piglets as a function of increasing standardized ileal digestible (SID) Arg: Lys in gestating sows. The optimum SID Arg: Lys ratio determined by straight broken-line model was 1.25 [Y = 17.1289 − 9.1486 (1.25 − SID Arg: Lys ratio) (closed line); R^2^ = 0.92687, *p* < 0.05]. Using the quadratic broken-line model, the optimum SID Arg: Lys ratio was 1.44, [Y = 17.3295 − 12.0702 × (1.44 − SID Arg: Lys)^2^ (open line); R^2^ = 0.8593, *p* < 0.05]. Data points (●) represent treatment means (n = 35, 34, 35, 36, 33).

**Table 1 animals-14-02688-t001:** Composition and nutrient level of the diets for gestating sows (air-dry basis).

Item	SID Arg: Lys Ratio				
	0.91	1.02	1.14	1.25	1.38
Ingredient, g/kg					
Corn	565.8	566.6	567.4	568.3	569.1
Wheat	100.0	100.0	100.0	100.0	100.0
Soybean meal, 44%	134.0	134.0	134.0	134.0	134.0
Alfalfa meal	128.0	128.0	128.0	128.0	128.0
Soybean oil	15.0	15.0	15.0	15.0	15.0
Dicalcium phosphate	9.3	9.3	9.3	9.3	9.3
Limestone	5.3	5.3	5.3	5.3	5.3
Salt	4.0	4.0	4.0	4.0	4.0
L-Lysine-HCl	2.3	2.3	2.3	2.3	2.3
L-Arg	0.0	0.8	1.6	2.4	3.2
L-Alanine	6.5	4.9	3.3	1.6	0.0
DL-Methionine	1.0	1.0	1.0	1.0	1.0
L-Threonine	1.0	1.0	1.0	1.0	1.0
Premix ^a^	30.0	30.0	30.0	30.0	3.0
Total	1000.0	1000.0	1000.0	1000.0	1000.0
Analyzed nutrient level					
Crude protein, %	14.24	14.25	14.26	14.28	14.26
Arg	0.64	0.71	0.79	0.89	0.97
Lysine	0.79	0.79	0.80	0.80	0.78
Calculated composition ^b^					
Digestible energy, Mcal/kg	3.14	3.14	3.14	3.14	3.14
Net energy, Mcal/kg	2.30	2.30	2.30	2.30	2.30
Crude fiber, %	6.02	6.02	6.02	6.02	6.02
Calcium, %	0.82	0.82	0.82	0.82	0.82
Phosphorus, %	0.48	0.48	0.48	0.48	0.48
STTD Phosphoru, %	0.35	0.35	0.35	0.35	0.35
SID Arg, %	0.58	0.65	0.73	0.80	0.88
SID Lysine, %	0.64	0.64	0.64	0.64	0.64
SID Arg: Lys, %	0.91	1.02	1.14	1.25	1.38
SID Methionine + Cysteine, %	0.41	0.41	0.41	0.41	0.41
SID Threonine, %	0.42	0.42	0.42	0.42	0.42
SID Tryptophan, %	0.14	0.14	0.14	0.14	0.14

^a^ Provide the following per kg complete diet: Vitamin A, 12,000 IU; Vitamin D_3_, 2400 IU; Vitamin E, 44 IU; Vitamin K, 2 mg; Biotin, 1 mg; folic acid, 5 mg; Vitamin B_1_, 4 mg; Vitamin B_2_, 15 mg; Vitamin B_6_, 5 mg; Vitamin B_12_, 50 μg; Niacin, 30 mg; Pantothenic acid, 40 mg; Choline chloride, 800 mg; Fe (ferrous sulfate), 100 mg; Cu (copper sulfate), 12 mg; Mn (manganese sulfate), 25 mg; Zn (zinc sulfate), 120 mg; I (calcium iodate), 0.2 mg; Se (sodium selenite), 0.2 mg. ^b^ Calculated chemical concentrations using nutritional values for feed ingredients from the NRC (2012). STTD: standardized total tract digestibility. SID: standardized ileal digestible.

**Table 2 animals-14-02688-t002:** Effect of dietary standardized ileal digestible Arg: Lys ratio during gestation on reproductive performance of sows.

Item	SID Arg: Lys Ratio					SEM	*p*-Value		
	0.91	1.02	1.14	1.25	1.38		Treatment	Linear	Quadratic
Sow number, n	35	34	35	36	33				
Litter size, n									
Total born	12.23	12.94	12.94	13.03	12.61	0.25	0.843	0.648	0.496
Born alive	10.14 ^c^	10.67 ^bc^	11.46 ^b^	12.50 ^a^	12.03 ^ab^	0.23	0.005	0.001	0.001
Stillborn	1.57 ^a^	1.48 ^a^	0.89 ^ab^	0.14 ^c^	0.27 ^bc^	0.11	0.000	0.000	0.000
Mummified	0.09	0.30	0.23	0.14	0.03	0.03	0.061	0.236	0.033
Average live birth weight, kg	1.41	1.40	1.41	1.41	1.41	0.01	0.992	0.846	0.980
Litter weight of born alive, kg	14.30 ^b^	14.49 ^b^	16.15 ^ab^	17.62 ^a^	16.96 ^a^	0.33	0.003	0.001	0.001
CV of total born birth weitght	0.21 ^a^	0.20 ^ab^	0.18 ^ab^	0.17 ^b^	0.16 ^b^	0.03	0.020	0.002	0.008
Farrowing duration, min	283 ^a^	220 ^b^	234 ^b^	259 ^ab^	223 ^b^	7.31	0.030	0.115	0.192
Birth intervals, min	25.86 ^a^	18.22 ^b^	18.61 ^b^	19.87 ^b^	18.51 ^b^	0.82	0.014	0.025	0.013
Backfat thickness, mm									
Day 30 of gestation	17.17	16.76	16.83	16.94	16.76	0.25	0.984	0.718	0.985
Day 110 of gestation	20.14	19.45	19.80	20.03	19.67	0.26	0.925	0.833	0.959
Backfat gain (d 30 to 110)	2.97	2.70	2.97	3.08	2.91	0.06	0.268	0.515	0.808

SEM: standard error of the mean, SID: standardized ileal digestibility. ^a–c^ Means within a row with different superscripts indicate significant differences (*p* < 0.05).

**Table 3 animals-14-02688-t003:** Effect of dietary standardized ileal digestible Arg: Lys ratio during gestation on the composition of colostrum in sows.

Item	SID Arg: Lys Ratio					SEM	*p*-Value		
	0.91	1.02	1.14	1.25	1.38		Treatment	Linear	Quadratic
Milk fat, %	2.62	3.61	3.78	3.99	3.45	0.17	0.109	0.109	0.023
Lactose, %	4.24 ^a^	3.62 ^b^	4.02 ^ab^	3.64 ^b^	4.01 ^ab^	0.08	0.041	0.492	0.153
Non-fat solid, %	17.94 ^c^	20.20 ^b^	18.38 ^c^	22.34 ^a^	20.13 ^b^	0.38	<0.001	0.017	0.029
Total-solid, %	20.63 ^c^	24.12 ^b^	22.44 ^bc^	26.83 ^a^	23.96 ^b^	0.48	<0.001	0.006	0.003
Milk protein, %	12.01 ^d^	14.93 ^b^	12.68 ^cd^	16.83 ^a^	14.36 ^bc^	0.40	<0.001	0.023	0.025
Energy, kJ/100 g	367 ^c^	419 ^b^	384 ^c^	478 ^a^	423 ^b^	18.99	<0.001	0.015	0.017

SEM, standard error of the mean (n = 6). ^a–d^ Means within a row with different superscripts indicate significant differences (*p* < 0.05). Energy, kJ/100 g = 39.8 (kJ/g) × fat (%) + 23.9 (kJ/g) × protein (%) + 16.5 (kJ/g) × lactose (%).

**Table 4 animals-14-02688-t004:** Effect of dietary standardized ileal digestible Arg: Lys ratio during gestation on plasma IgA, IgG, IgM in sows.

Item	SID Arg: Lys Ratio					SEM	*p*-Value		
	0.91	1.02	1.14	1.25	1.38		Treatment	Linear	Quadratic
IgA μg/mL									
90 d	8.77	14.34	13.06	15.34	11.59	0.48	0.060	0.148	0.028
110 d	10.85	12.52	11.65	12.08	11.22	0.72	0.187	0.450	0.581
IgG μg/mL									
90 d	81.52	105.27	98.42	89.84	88.62	1.56	0.235	0.145	0.169
110 d	79.42	89.85	86.43	110.32	95.98	1.65	0.090	0.011	0.014
IgM μg/mL									
90 d	41.19	43.68	38.90	46.08	52.41	1.72	0.109	0.037	0.042
110 d	43.18	44.72	49.53	46.24	47.26	0.91	0.178	0.119	0.132

SEM, standard error of the mean (n = 6).

**Table 5 animals-14-02688-t005:** Effect of dietary standardized ileal digestible Arg: Lys on the plasma urea nitrogen and NO concentrations in gestating sows.

Item	SID Arg: Lys Ratio					SEM	*p*-Value		
	0.91	1.02	1.14	1.25	1.38		Treatment	Linear	Quadratic
Urea nitrogen, mmol/L									
Day 90	6.28	6.58	5.56	5.75	6.14	0.23	0.645	0.572	0.600
Day 110	6.33	6.05	5.75	6.12	5.58	0.16	0.622	0.204	0.450
NO, μmol/L									
Day 90	8.89	8.89	7.86	8.37	8.93	0.25	0.701	0.496	0.531
Day 110	7.56 ^b^	8.56 ^a^	8.51 ^a^	8.78 ^a^	9.33 ^a^	0.17	0.014	0.001	0.004
IGF-1, ug/L									
Day 90	29.98	31.27	30.01	30.61	32.14	0.49	0.613	0.280	0.483
Day 110	30.49	30.76	30.68	30.1	30.28	0.20	0.850	0.468	0.725
Progesterone, ng/ml									
Day 90	343.77	388.84	341.80	369.47	414.80	13.63	0.409	0.196	0.358
Day 110	372.04	360.00	382.81	345.45	354.60	8.44	0.688	0.420	0.714
GH, ng/ml									
Day 90	0.49	0.55	0.49	0.53	0.53	0.02	0.287	0.084	0.141
Day 110	0.54	0.53	0.56	0.51	0.55	0.01	0.843	0.706	0.930
Insulin, mIU/L									
Day 90	12.71 ^b^	12.79 ^b^	12.31 ^b^	18.19 ^a^	17.08 ^a^	0.91	0.028	0.007	0.025
Day 110	15.37	17.02	15.99	18.86	15.59	0.84	0.702	0.730	0.640
Estrogen, pmol/L									
Day 90	7.43	10.02	7.44	8.55	10.99	0.74	0.472	0.270	0.478
Day 110	7.18	8.73	9.37	8.42	9.72	0.63	0.771	0.287	0.534

SEM, standard error of the mean (n = 6). ^a,b^ Means within a row with different superscripts indicate significant differences (*p* < 0.05).

**Table 6 animals-14-02688-t006:** Effect of dietary standardized ileal digestible Arg: Lys ratio on concentrations of plasma biochemical indices in sows at day 110 of gestation.

Item	SID Arg: Lys Ratio					SEM	*p*-Value		
	0.91	1.02	1.14	1.25	1.38		Treatment	Linear	Quadratic
Total protein, g/L	71.98	73.56	76.57	74.09	80.30	3.76	0.233	0.042	0.123
Albumin, g/L	39.08	36.27	36.65	38.12	41.25	2.54	0.336	0.275	0.094
Creatinine, μmol/L	174	184	188	197	183	12.00	0.430	0.249	0.186
Uric acid, μmol/L	36.17	39.55	43.28	45.36	51.31	10.08	0.628	0.100	0.264
Glucose, mmol/L	4.92	5.38	5.61	5.36	4.87	0.56	0.62	0.894	0.259
Triacylglycerol, mmol/L	2.94	3.68	4.28	3.20	3.04	0.74	0.050	0.379	0.215
Cholesterol, mmol/L	1.33	1.13	1.37	1.31	1.40	0.14	0.346	0.304	0.463
HDL-Cholesterol, mmol/L	4.83	4.62	5.50	5.47	5.49	0.58	0.390	0.096	0.238
LDL-Cholesterol, mmol/L	0.51	0.44	0.54	0.51	0.55	0.48	0.236	0.186	0.360
Calcium, mmol/L	2.89	3.03	3.13	3.16	3.34	0.256	0.521	0.069	0.198
Phosphorus, mmol/L	1.69	1.76	1.57	1.67	1.80	0.091	0.126	0.524	0.196

SEM: standard error of the mean (n = 6), HDL: high-density lipoprotein, LDL: low-density lipoprotein.

**Table 7 animals-14-02688-t007:** Effect of dietary standardized ileal digestible Arg: Lys ratio on the plasma concentrations of free amino acids in sows at day 90 of gestation.

Item/μmol/L	SID Arg: Lys Ratio					SEM	*p*-Value		
	0.91	1.02	1.14	1.25	1.38		Treatment	Linear	Quadratic
Taurine	69	63	69	62	64	1.97	0.697	0.474	0.744
Aspartate	14	14	12	13	11	0.70	0.593	0.118	0.277
Threonine	204 ^b^	193 ^b^	205 ^b^	163 ^b^	262 ^a^	9.13	0.005	0.156	0.023
Serine	173	162	159	160	172	4.19	0.718	0.902	0.339
Asparagine	74	70	72	75	83	4.03	0.907	0.465	0.592
Glutamate	171 ^a^	130 ^ab^	151 ^a^	145 ^ab^	103 ^b^	7.63	0.053	0.021	0.064
Glutamine	386	374	348	379	330	11.45	0.517	0.177	0.402
Glycine	964	863	884	917	1006	39.04	0.795	0.607	0.448
Alanine	990 ^a^	820 ^ab^	716 ^b^	726 ^b^	666 ^b^	33.29	0.008	0.001	0.001
Citrulline	100	90	101	122	123	6.29	0.399	0.082	0.199
Valine	299	281	302	282	337	9.38	0.330	0.220	0.194
Cystine	15	15	14	12	15	0.84	0.801	0.748	0.744
Methionine	75 ^ab^	87 ^a^	83 ^a^	63 ^b^	90 ^a^	2.63	0.023	0.727	0.744
Isoleucine	139	140	139	134	157	4.71	0.605	0.375	0.379
Leucine	279	263	250	257	289	8.33	0.584	0.786	0.237
Tyrosine	145 ^b^	131 ^b^	136 ^b^	145 ^b^	169 ^a^	4.19	0.031	0.031	0.004
Phenylalanine	106	102	113	107	121	2.73	0.193	0.061	0.126
Ornithine	113 ^b^	135 ^b^	131 ^b^	174 ^a^	175 ^a^	6.12	<0.001	<0.001	<0.001
Lysine	187	203	202	178	210	6.12	0.488	0.620	0.864
Histidine	121	109	117	120	117	2.80	0.738	0.839	0.887
Arg	180 ^b^	234 ^b^	237 ^b^	323 ^a^	327 ^a^	14.63	0.001	<0.001	<0.001
Proline	309	309	297	322	333	11.73	0.911	0.467	0.665

SEM, standard error of the mean (n = 6). ^a,b^ Means within a row with different superscripts indicate significant differences (*p* < 0.05).

**Table 8 animals-14-02688-t008:** Effect of dietary standardized ileal digestible Arg: Lys ratio on the plasma concentrations of free amino acids in sows at day 110 of gestation. μmol/L.

Item/μmol/L	SID Arg: Lys Ratio					SEM	*p*-Value		
	0.91	1.02	1.14	1.25	1.38		Treatment	Linear	Quadratic
Taurine	48	55	54	58	63	0.83	0.115	0.008	0.033
Aspartate	11	13	12	12	11	0.70	0.138	0.298	0.132
Threonine	204	202	201	232	206	6.73	0.576	0.496	0.733
Serine	142	168	164	179	167	5.70	0.361	0.145	0.163
Asparagine	71	105	115	108	93	6.39	0.215	0.315	0.055
Glutamate	138	146	158	144	139	4.53	0.649	0.965	0.400
Glutamine	254 ^b^	335 ^a^	335 ^a^	385 ^a^	328 ^ab^	13.84	0.041	0.047	0.014
Glycine	534	701	563	647	590	25.37	0.234	0.787	0.580
Alanine	1016 ^a^	999 ^a^	881 ^ab^	804 ^b^	740 ^b^	29.54	0.003	<0.001	<0.001
Citrulline	70	70	66	83	61	2.69	0.121	0.733	0.467
Valine	246	282	288	318	290	10.75	0.345	0.110	0.133
Cystine	12	13	9	15	13	0.92	0.375	0.653	0.842
Methionine	79 ^b^	93 ^ab^	110 ^a^	118 ^a^	111 ^a^	4.79	0.050	0.006	0.008
Isoleucine	115	147	150	159	158	5.45	0.051	0.009	0.011
Leucine	234	309	291	317	289	10.29	0.082	0.117	0.043
Tyrosine	133	139	145	152	143	5.49	0.862	0.389	0.562
Phenylalanine	100	110	111	124	111	3.95	0.459	0.205	0.250
Ornithine	159	162	176	186	179	5.30	0.215	0.006	0.298
Lysine	140 ^c^	183 ^bc^	207 ^abc^	242 ^a^	227 ^ab^	9.73	0.003	<0.001	<0.001
Histidine	116	118	121	124	109	3.02	0.645	0.649	0.395
Arg	202 ^b^	206 ^b^	232 ^ab^	283 ^a^	299 ^a^	12.51	0.023	0.001	0.004
Proline	356	391	391	379	364	18.86	0.097	0.993	0.797

SEM, standard error of the mean (n = 6).^a–c^ Means within a row with different superscripts indicate significant differences (*p* < 0.05).

## Data Availability

All the datasets used and analyzed during the current study are included in the manuscript.

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
