# Peer review of "Effect of Dietary Standardized Ileal Digestible Arginine to Lysine Ratio on Reproductive Performance, Plasma Biochemical Index, and Immunity of Gestating Sows"

_animals, 2024, doi:10.3390/ani14182688_

Round 1
Reviewer 1 Report
Comments and Suggestions for Authors
In this study, Wen et al found that increasing the dietary SID Arg: Lys ratio during gestation resulted in the increase of litter size alive, reduction of birth intervals, and improvement of immunity and colostrum composition in sows. They suggested that the optimal SID Arg: Lys ratio for gestating sows to maximize litter size alive was 1.25. These results are very important for the improvement of sow reproduction performance. As the authors discussed, different amount of arginine and SID Arg: Lys ratios have different effects on lysine absorption. Could the author conclude that among which amount of arginine and SID Arg: Lys ratios in the diet, the absorption of lysine will be increased. Minor concern: line 71, threonine (averaged SID threonine 42%).
Author Response
Comments 1
As the authors discussed, different amount of arginine and SID Arg: Lys ratios have different effects on lysine absorption. Could the author conclude that among which amount of arginine and SID Arg: Lys ratios in the diet, the absorption of lysine will be increased.
Thank you for your comments .In the present study, we discovered that increasing the dietary arginine level and arginine to lysine ratio within a certain range significantly increased the plasma concentration of lysine on day 110 of gestation. The plasma lysine concentration was highest at a dietary arginine level of 0.80 and an arginine to lysine ratio of 1.25; therefore, we can infer that the efficiency of lysine absorption was highest at a dietary arginine level of 0.80 and an arginine to lysine ratio
Comments 2
line 71, threonine (averaged SID threonine 42%)
Thanks for pointing this out, I have revised it.
Reviewer 2 Report
Comments and Suggestions for Authors
This study mainly investigated the optimal standardized ileal digestible arginine to lysine (SID Arg: Lys) ratio on the reproductive performance, immunity and biochemical parameters of sows during gestation, as well as colostrum compositions and performance of their offspring litter. The results showed that the optimal SID Arg: Lys ratio for gestating sows to maximize litter size alive was 1.25.
There are some points in need of revision:
1. Please explain the purpose of detecting the insulin content in the plasma of the sows.
2. In the discussion section, please discuss the concentration of NO in the plasma of sows at day 110 of gestation in detail.
3. Some stylistic style modifications and moderate language improvement are needed.
4. Line 22, “The broken-line” revise to “Broken line”.
5. Line 30-31, “d” revise to “day”.
6. Line 63, “conducted in” revise to “performed on”.
7. Line 75, “moving to” revise to “they were moved to”.
8. Line 86, “with” revise to “using”.
9. Line 115, “determine” revise to “measure”.
10. Line 116, The proper noun abbreviation must be written after the full name. Change “IGF-1” to “Insulin-Like Growth Factor-1 (IGF-1).”
11. Line 123, “concentrations” revise to “concentration”.
12. Line 138-139, “At the same time, CV of total born birth weight was linearly (P < 0.01) and quadratically (P < 0.01) reduced by increasing the dietary SID Arg: Lys ratio” revise to “The CV of total born birth weight decreased linearly (P < 0.01) and quadratically (P < 0.01) as the dietary SID Arg:Lys ratio increased.”
13. Line 181, “composition” revise to “the composition”.
14. Line 203, “90th” revise to “day 90”.
15. Line 228-229, “There were no differences in concentrations of other free amino acids in plasma between groups (P > 0.05).” revise to “There were no differences in the concentrations of the other free amino acids in plasma among groups (P > 0.05)”.
16. Line 302-303, “The result indicated that the appropriate Arg: Lys ratio is conducive to the improvement of immune performance.” revise to “This finding indicates that an appropriate Arg:Lys ratio is critically important for improving immune performance.”
17. Line 323, “may” revise to “might”.
Author Response
Comments 1
- Please explain the purpose of detecting the insulin content in the plasma of the sows.
Thanks for pointing this out, we agree with this comments, I have provide the necessary explanation. The revised sections can be found in the manuscript.
Comments 2
- In the discussion section, please discuss the concentration of NO in the plasma of sows at day 110 of gestation in detail.
Thanks for pointing this out, we agree with this comments, I have discussed the changes of NO in the plasma. The revised sections can be found in the manuscript.
Comments 3
- Some stylistic style modifications and moderate language improvement are needed.
Thanks for pointing this out, we agree with this comments, I have revised it.
Comments 4
- Line 22, “The broken-line” revise to “Broken line”.
Thanks for pointing this out, we agree with this comments, I have revised it.
Comments 5
- Line 30-31, “d” revise to “day”.
Thanks for pointing this out, we agree with this comments, I have revised it.
Comments 6
- Line 63, “conducted in” revise to “performed on”.
Thanks for pointing this out, we agree with this comments, I have revised it.
Comments 7
- Line 75, “moving to” revise to “they were moved to”.
Thanks for pointing this out, we agree with this comments, I have revised it.
Comments 8
- Line 86, “with” revise to “using”.
Thanks for pointing this out, we agree with this comments, I have revised it.
Comments 9
- Line 115, “determine” revise to “measure”.
Thanks for pointing this out, we agree with this comments, I have revised it.
Comments 10
- Line 116, The proper noun abbreviation must be written after the full name. Change “IGF-1” to “Insulin-Like Growth Factor-1 (IGF-1).”
Thanks for pointing this out, we agree with this comments, I have revised it.
Comments 1
- Line 123, “concentrations” revise to “concentration”.
Thanks for pointing this out, we agree with this comments, I have revised it.
Comments 12
- Line 138-139, “At the same time, CV of total born birth weight was linearly (P < 0.01) and quadratically (P < 0.01) reduced by increasing the dietary SID Arg: Lys ratio” revise to “The CV of total born birth weight decreased linearly (P< 0.01) and quadratically (P < 0.01) as the dietary SID Arg:Lys ratio increased.”
Thanks for pointing this out, we agree with this comments, I have revised it.
Comments 13
- Line 181, “composition” revise to “the composition”.
Thanks for pointing this out, we agree with this comments, I have revised it.
Comments 14
- Line 203, “90th” revise to “day 90”.
Thanks for pointing this out, we agree with this comments, I have revised it.
Comments 15
- Line 228-229, “There were no differences in concentrations of other free amino acids in plasma between groups (P> 0.05).” revise to “There were no differences in the concentrations of the other free amino acids in plasma among groups (P > 0.05)”.
Thanks for pointing this out, we agree with this comments, I have revised it.
Comments 16
- Line 302-303, “The result indicated that the appropriate Arg: Lys ratio is conducive to the improvement of immune performance.” revise to “This finding indicates that an appropriate Arg:Lys ratio is critically important for improving immune performance.”
Thanks for pointing this out, we agree with this comments, I have revised it.
Comments 17
- Line 323, “may” revise to “might”.
Thanks for pointing this out, we agree with this comments, I have revised it.
Reviewer 3 Report
Comments and Suggestions for Authors
This manuscript has investigated the effect of Arg to Lys ratio for gestating sows. Correct terms need to be used (litter size alive – the number of born alive, etc.). Please check English. Also, this manuscript needs to justify the validity of broken-line analysis.
L12 please add simple summary.
L21 reduced with increasing
L23 the number of born alive (not litter size alive) throughout the manuscript.
L26-27 please specify ‘plasma’ of ‘sows’
L28 please revise to ‘decrease’ or ‘increase’ instead of significant effects
L51 2.03 is not correct. In the reference (7), is it supposed to be 1.03 from Table 6?
L60-62 This part is not needed. IACUC statement is needed.
L64 Was this study started at mating? Please indicate this.
L80 Please define abbreviations used in the tables.
L83 what are “Ca:32.5g; P:8.75g” for?
L84 Please add amino acid analysis to confirm the arginine content and ratio to Lys.
L88 The number of total born, born alive…
L96 indices and…
L99 Is there any specific reason for not collecting milk samples during lactation other than colostrum?
L104 Ma et al.
L107 please define UN and NO
L120 sow or litter
L119-131 Why SAS version is different?
L141-142 Due to only one point (SID Arg:Lys; 1.38) above estimated optimal SID Arg:Lys ratio, this value (1.25) is not valid since we can’t know what will happen in the ratio over 1.38. If there is an increase it could be linear. In order to confirm this is plateaued over 1.25, higher SID Arg:Lys ratio is needed. The authors can still say 1.25 maximized the litter size and litter weight but it is not guaranteed to say 1.25 is optimal without higher ratio tested.
L191 Can colostrum IgG, A be analyzed?
Please define abbreviations in the tables.
L220 tyramine? Missing methionine, tyrosine at d 90
L224 glutamine instead of glutamate? (please check p-values, P<0.05)
Please double check Table 7 and 8 results.
L246 This paper needs to justify the validity of broken-line analysis.
L250 please add NRC SID Arg:Lys ratio values here again.
L255 please be consistent with Arg or arginine
L263 ‘by improvement of body condition of sows by arginine’ needs a reference
L264 promoted
L266-267 this sentence is not valid. Please discuss further regarding why there is a discrepancy between studies. What kind of treatment condition, animal species, processing time and composition of diets were different? How were those impacted to results?
L301 how increased Ig in plasma of sows is correlated with colostrum Ig? As the colostrum samples were collected in the study, having colostrum Ig values should improve the strength of this paper.
L332 delete promotes?
Table 8 Isoleucine – please check p-value for treatment. There should be differences.
L341-343 again this sentence is not valid. There is discrepancy between the current study and previous study in lysine in plasma. There should be more discussion instead of saying the ratio and amount of Arg are different. How were they different? What are the impact of it?
Comments on the Quality of English LanguagePlease see the comments.
Author Response
This manuscript has investigated the effect of Arg to Lys ratio for gestating sows. Correct terms need to be used (litter size alive – the number of born alive, etc.). Please check English. Also, this manuscript needs to justify the validity of broken-line analysis.
Response: Thank you for pointing this out, I have revised it.
L12 please add simple summary.
Response: Thank you for pointing this out, I have added simple summary.
L21 reduced with increasing
Response: Thank you for pointing this out, we agree with this comment, I have revised it
L23 the number of born alive (not litter size alive) throughout the manuscript.
Response: Thank you for pointing this out, I have revised it throughout the manuscript.
L26-27 please specify ‘plasma’ of ‘sows’
Response: Thank you for pointing this out, I have revised it in manuscript.
Response: L28 please revise to ‘decrease’ or ‘increase’ instead of significant effects
Thank you for pointing this out, I have revised it in manuscript.
L51 2.03 is not correct. In the reference (7), is it supposed to be 1.03 from Table 6?
Response: Yes, It is calculated from Table 6. I have revised it in manuscript.
L60-62 This part is not needed. IACUC statement is needed.
Response: Thank you for pointing this out, I have delete this part and added IACUC statement.
L64 Was this study started at mating? Please indicate this.
Response: Thank you for pointing this out, this study started at day 30 of gestation.
L80 Please define abbreviations used in the tables.
Response: Thank you for pointing this out, I have revised it in manuscript.
L83 what are “Ca:32.5g; P:8.75g” for?
Response: The premix contained CaCO3 and CaH2PO4.
L84 Please add amino acid analysis to confirm the arginine content and ratio to Lys.
Response: Thank you for pointing this out. I have added analyzed nutrient level of lysine and arginine.
L88 The number of total born, born alive…
Response: Thank you for pointing this out, I have revised it in manuscript.
L96 indices and…
Response: Thank you for pointing this out, I have revised it in manuscript.
L99 Is there any specific reason for not collecting milk samples during lactation other than colostrum?
Response: Because this study focused on the reproductive performance of pregnancy, the milk data during lactation was not included.
L104 Ma et al.
Response: Thank you for pointing this out, I have revised it in manuscript.
L107 please define UN and NO
Response: Thank you for pointing this out, I have revised it in manuscript.
L120 sow or litter
Response: Sow, thank you for pointing this out, I have revised it in manuscript
L119-131 Why SAS version is different?
Response: Both are version 9.2, I have revised it in manuscript
L141-142 Due to only one point (SID Arg:Lys; 1.38) above estimated optimal SID Arg:Lys ratio, this value (1.25) is not valid since we can’t know what will happen in the ratio over 1.38. If there is an increase it could be linear. In order to confirm this is plateaued over 1.25, higher SID Arg:Lys ratio is needed. The authors can still say 1.25 maximized the litter size and litter weight but it is not guaranteed to say 1.25 is optimal without higher ratio tested.
Response: Thank you for pointing this out. If the arginine to lysine ratio continues to increase, the litter size and litter weight may also increase. I have revised it in manuscript.
L191 Can colostrum IgG, A be analyzed?
Response: Thank you for pointing this out. In the following experiments I will test for immunoglobulin in colostrum.
Please define abbreviations in the tables.
Response: Thank you for pointing this out, I have defined the abbreviations of IgA, IgG, IgM.
L220 tyramine? Missing methionine, tyrosine at d 90
Response: Tyrosine, I have revised it in manuscript. Added description of methionine results
L224 glutamine instead of glutamate? (please check p-values, P<0.05)
Response: Thank you for pointing this out, it is glutamine, have revised it in manuscript.
Please double check Table 7 and 8 results.
Response: Thank you for your suggestion. I have checked Table 7 and Table 8 carefully. And made a modification.
L246 This paper needs to justify the validity of broken-line analysis.
Response: Thank you for pointing this out, I have added P-value.
L250 please add NRC SID Arg:Lys ratio values here again.
Response: NRC SID Arg:Lys ratio was 0.53, I have added it.
L255 please be consistent with Arg or arginine
Response: Thank you for pointing this out, I have revised it in manuscript.
L263 ‘by improvement of body condition of sows by arginine’ needs a reference
Response: Thank you for pointing this out, I have revised it in manuscript.
L264 promoted
Response: Thank you for pointing this out, I have revised it in manuscript.
L266-267 this sentence is not valid. Please discuss further regarding why there is a discrepancy between studies. What kind of treatment condition, animal species, processing time and composition of diets were different? How were those impacted to results?
Response: Thank you for pointing this out. I have removed this sentence and added the discussion in this section.
L301 how increased Ig in plasma of sows is correlated with colostrum Ig? As the colostrum samples were collected in the study, having colostrum Ig values should improve the strength of this paper.
Response: Thank you for pointing this out. In the following experiments I will test for immunoglobulin in colostrum.
L332 delete promotes?
Response: Thank you for pointing this out, I have delete it in manuscript.
Table 8 Isoleucine – please check p-value for treatment. There should be differences.
Response: Thank you for pointing this out, p-value was 0.051. There is a trend of influence.
L341-343 again this sentence is not valid. There is discrepancy between the current study and previous study in lysine in plasma. There should be more discussion instead of saying the ratio and amount of Arg are different. How were they different? What are the impact of it?
Response: Thank you for pointing this out. I have removed this sentence and added the discussion in this section.
Round 2
Reviewer 3 Report
Comments and Suggestions for Authors
“L83 what are “Ca:32.5g; P:8.75g” for?
Response: The premix contained CaCO3 and CaH2PO4.”
Authors may remove this Ca and P in the premix.
“L141-142 Due to only one point (SID Arg:Lys; 1.38) above estimated optimal SID Arg:Lys ratio, this value (1.25) is not valid since we can’t know what will happen in the ratio over 1.38. If there is an increase it could be linear. In order to confirm this is plateaued over 1.25, higher SID Arg:Lys ratio is needed. The authors can still say 1.25 maximized the litter size and litter weight but it is not guaranteed to say 1.25 is optimal without higher ratio tested.”
“Response: Thank you for pointing this out. If the arginine to lysine ratio continues to increase, the litter size and litter weight may also increase. I have revised it in manuscript.”
In this Revision 1, authors did not revise well enough to be acceptable. If litter size and weight may increase with further increase of SID Arg:Lys, it can not be concluded that 1.25 is the optimal. Please add discussion about the limitation of this study for determining 1.25 as an optimal ratio.
L268 – what is the unit of 0.28?
Still need to revise for Arg or arginine
L284 needs references for these ‘other studies with no differences’
L366 please add references regarding the high level of arginine reduce lysine absorption.
Comments on the Quality of English Language
None.
Author Response
Comments 1
“L83 what are “Ca: 32.5g; P:8.75g” for?
Response: The premix contained CaCO3 and CaH2PO4.”
Authors may remove this Ca and P in the premix.
Response:Thank you for pointing this out. I have removed it in the premix, and added in Dicalcium phosphate and Limestone.
Comments 2
“L141-142 Due to only one point (SID Arg:Lys; 1.38) above estimated optimal SID Arg:Lys ratio, this value (1.25) is not valid since we can’t know what will happen in the ratio over 1.38. If there is an increase it could be linear. In order to confirm this is plateaued over 1.25, higher SID Arg:Lys ratio is needed. The authors can still say 1.25 maximized the litter size and litter weight but it is not guaranteed to say 1.25 is optimal without higher ratio tested.”
“Response: Thank you for pointing this out. If the arginine to lysine ratio continues to increase, the litter size and litter weight may also increase. I have revised it in manuscript.”
In this Revision 1, authors did not revise well enough to be acceptable. If litter size and weight may increase with further increase of SID Arg:Lys, it can not be concluded that 1.25 is the optimal. Please add discussion about the limitation of this study for determining 1.25 as an optimal ratio.
Response:Thank you for pointing this out. I have added in discussion.
Comments 3
L268 – what is the unit of 0.28?
Still need to revise for Arg or arginine
Response:Thank you for pointing this out. The unit is %, and added it.
Comments 4
L284 needs references for these ‘other studies with no differences’
Response:Thank you for pointing this out. I have added a reference.
Comments 5
L366 please add references regarding the high level of arginine reduce lysine absorption.
Response:Thank you for pointing this out. I have added a reference.